# The Mediating Effect of Basic Psychological Needs Satisfaction between Future Socio-Economic Status and Undergraduates' Sense of Meaning in Life

Feng Zhang [1,2], Li Su [1] and Xiaowei Geng [1,3,*]

1 Jing Hengyi School of Education, Hangzhou Normal University, Hangzhou 311121, China
2 Chinese Education Modernization Research Institute, Hangzhou Normal University (Zhejiang Provincial Key Think Tank), Hangzhou 311121, China
3 Zhejiang Philosophy and Social Science Laboratory for Research in Early Development and Childcare, Hangzhou Normal University, Hangzhou 311121, China
* Correspondence: xwgeng@hznu.edu.cn

**Abstract:** Finding meaning in life helps improve undergraduates' well-being, hope, and adaptation to life. To investigate the relationship between future socioeconomic status (future SES) and undergraduates' sense of meaning in life, 333 undergraduates were surveyed using the Meaning in Life Scale, Basic Psychological Needs Scales, and Social Ladder Scale. Results showed that (1) undergraduates' sense of meaning in life was at the middle level, their current SES was low, but their future SES was high. (2) The sense of meaning in life was significantly predicted by future SES, with sex, age, objective SES, and current SES controlled. (3) Basic psychological needs satisfaction mediated the influence of future SES on sense of meaning in life. Thus, undergraduates expected future SES influenced their sense of meaning in life through satisfying their basic psychological needs, i.e., relatedness need, autonomy need, and competence need.

**Keywords:** future socioeconomic status; sense of meaning in life; basic psychological needs satisfaction; undergraduates



## 1. Introduction

With the increasing social pressure and temptation of the virtual world (e.g., online games), the sense of meaning in life among undergraduates in China is not optimistic (Zhang and Zhou 2020; Gao et al. 2017). Studies have shown that more than 50% of undergraduates' life goals are not clear, and their sense of meaning in life is low (Li and Lu 2010; Ding et al. 2016; Zhou and Wu 2021; Wang et al. 2018; Hu et al. 2020; Xie 2021). Meaning in life plays an important role in the development of an individual's physical and mental health and the improvement of their confidence in their future (Hou et al. 2021; Wang et al. 2016). Therefore, studying how to improve the sense of meaning in life for undergraduates is of great importance.

Meaning in life is defined as the feeling that life is meaningful and valuable and has a clear goal, mission, or purpose (Steger 2009), which consists of coherence, purpose, and significance. Coherence means a sense of comprehensibility and that one's life makes sense. Purpose means a sense of core goals, aims, and direction in life. Significance is about a sense of life's inherent value and having a life worth living (Martela and Steger 2016). Previous studies showed that the influencing factors of undergraduates' sense of meaning in life include biological sex, age, left-behind experience, family structure and parenting style, self-reflection, mindfulness, gratitude, etc. (Yu et al. 2019; Li 2019; Wang 2019; Duan 2019). However, few studies shed light on the effect of socio-economic status (SES), especially future SES, on meaning in the lives of undergraduates.

Socioeconomic status (SES) is a social classification used to reflect the relative position of individuals in the social hierarchy ladder, which includes subjective and objective SES

(Kraus et al. 2012; Hu et al. 2014). Subjective SES is the subjective perception of people's social status, which is usually measured by the social ladder scale (Adler et al. 2000). Objective SES represents the objective social resources occupied by him or her, including three aspects: income, education level, and occupational prestige. Individuals with higher SES have more material resources (e.g., higher income, higher education level, etc.) and social resources (e.g., friends, social relations, etc.). Thus, individuals with higher SES usually form independent selves, while those with lower SES usually form interdependent selves (Stephens et al. 2014). Previous studies showed that subjective family SES positively affects undergraduates' life meaning experience (Shang et al. 2016; Tang et al. 2021) and subjective well-being (Zhao et al. 2019). Research also showed that low SES led to worse psychological well-being through sleep stress and other somatic problems (Huynh and Chiang 2018). Moreover, the research on Hong Kong teenagers found that the perceived economic pressure significantly reduced their sense of meaning in life, which shows that SES has a great impact on the sense of meaning in life (Shek 2003).

Previous studies mostly focused on the current subjective SES and objective SES; few studies shed light on the future subjective SES. In the current research, to make it simpler to express, current subjective SES was abbreviated as current SES, and future subjective SES was abbreviated as future SES, which referred to the socioeconomic status people expect to be in the future. Undergraduates have not yet entered the workforce. After graduation, undergraduates will step into the workforce. Therefore, undergraduates usually have expectations for their future SES, and their future SES is usually not the same as their current SES. Studies suggested that imagining one's self in the future was also linked to meaning in life (King and Hicks 2021). Thus, we proposed that future SES would be positively related to undergraduates' sense of meaning in life.

Self-determination theory (SDT) contends that human beings have three basic psychological needs, i.e., autonomy need, competence need, and relatedness need (Deci and Ryan 2000). Competence needs refer to being successful at the most appropriate and challenging tasks and being able to achieve the desired results. Autonomy need refers to experiencing choice and feeling like a pioneer. Relatedness need refers to the establishment of a sense of mutual respect and dependence with others. Studies have shown that the satisfaction of the three basic psychological needs can improve an individual's happiness and sense of meaning (Weinstein et al. 2003; Martela et al. 2018; Zhang and Li 2018). If undergraduates expected that their future SES would be high, they would feel that they were more capable, have a stronger sense of autonomy, and get along well with the people around them. In other words, undergraduates with higher future SES can better meet the three basic psychological needs, which can improve their meaning in life. Thus, it is reasonable to expect that the satisfaction of basic psychological needs plays a mediating role in the impact of future SES on the sense of meaning in life.

In short, this study aims to examine whether and how undergraduates' future SES influences their sense of meaning in life. Based on the literature review, in the present research, we hypothesized that future SES positively predicts undergraduates' sense of meaning in life (Hypothesis 1). Basic psychological needs satisfaction would mediate the impact of future SES on undergraduates' sense of meaning in life (Hypothesis 2). Undergraduates with a higher expectation of future SES may meet their basic psychological needs better, which then further improves their sense of meaning in life.

## 2. Materials and Methods

### 2.1. Participants

A total of 333 undergraduates were selected from a university in China for convenience, including 189 males and 144 females. Their average age was 18.57 years, with a standard deviation of 0.78. Their major includes education, civil engineering, and law. A total of 71.4% of the participants came from rural areas, and 28.6% of them came from urban areas. The average education level of undergraduates' fathers is 1.78 (1 = junior high school and below, 2 = high school), and the average education level of their mothers is 1.59 (1 = junior

high school and below, 2 = high school), which means that the average education level of undergraduates' parents was high school and below.

*2.2. Measurements*

2.2.1. Current SES, Future SES, and Objective SES

Current and future SES were measured by the MacArthur Scale (Adler et al. 2000), which is widely used in studies of socioeconomic status (Kraus et al. 2009; Piff et al. 2010; Cheon and Hong 2017). It was also validated in the undergraduate sample (Piff et al. 2010; Piff et al. 2012; Geng et al. 2022). On this scale, subjects were presented with a 10-level ladder, which represents the social status of the individual. The top of the ladder represents people at the top of the socioeconomic status, who have the highest income, the highest level of education, or prestigious work. The bottom of the ladder represents people at the bottom of the socioeconomic status, who have the least income, the lowest education level, or are jobless. Undergraduates were required to choose which layer of the ladder they were on as an indicator of their current SES. In addition, they were asked to evaluate which layer of the social ladder they would stand on in the future as an indicator of their future SES.

Objective SES was also measured as the control variable, which includes three aspects: income, education level, and occupational prestige. In this study, we used monthly household income and educational level of parents as indicators to measure objective SES. The average monthly household income includes five categories: 1 = less than CNY 4000; 2 = CNY 4001–6000; 3 = CNY 6001–8000; 4 = CNY 8001–10000; 5 = more than CNY 10000; the educational level of father and mother includes five grades: 1 = junior high school and below; 2 = high school; 3 = junior college; 4 = undergraduate; 5 = master's degree and above. According to previous research (Kraus et al. 2009), the average monthly income of the family and the education level of parents were standardized first and then compromised as the objective SES indicators for undergraduates.

2.2.2. Sense of Meaning in Life

The Meaning in Life Questionnaire (MLQ), developed by Steger et al. (2006), was used to measure undergraduates' sense of meaning in life. The questionnaire includes two dimensions, i.e., meaning experience and meaning seeking. The meaning experience dimension includes 5 items, such as "I understand my life's meaning", and the meaning seeking dimension also includes 5 items, such as "I am always looking to find my life's purpose." Participants were asked to evaluate the degree of agreement (1 = strongly disagree, 7 = strongly agree). The higher the score, the stronger the sense of meaning in life. The internal consistency of Cronbach's α coefficient for the total score was 0.79.

2.2.3. Basic Psychological Needs Satisfaction

Basic psychological needs satisfaction was measured by Basic Psychological Needs Scales (BPNS) developed by Gagné in 2003 (Deci and Ryan 2008), which includes three subscales, i.e., competency need satisfaction, autonomy need satisfaction, and relatedness need satisfaction. Competency need satisfaction includes six items, for example, "People I know tell me I am good at what I do" and "I have been able to learn interesting new skills recently"; autonomy need satisfaction includes six items, e.g., "There is not much opportunity for me to decide for myself how to do things in my daily life" and "I feel like I can pretty much be myself in my daily situations"; relatedness need satisfaction includes seven items, such as "People in my life care about me" and "People are generally pretty friendly towards me". Each item was scored on a seven-point scale, where 1 = not true at all and 7 = definitely true. The higher the score, the higher the satisfaction of basic psychological needs. The internal consistency of Cronbach's α coefficient for the total score was 0.84.

*2.3. Analytic Approach*

Statistical analyses were conducted using SPSS 21.0 (the Statistical Package for the Social Sciences) which has been first developed by three PhD students at the University of Stanford and was sourced from Hangzhou, China. A *t*-test was used to test differences in the sense of meaning in life and the SES of undergraduates of different sexes and different areas. Correlation analysis tested the relationship between undergraduates' SES and sense of meaning in life. Regression analysis was used to test the mediating role of satisfaction of basic psychological needs in the relationship between future SES and undergraduates' sense of meaning in life.

**3. Results**

*3.1. Descriptive Statistics on the Meaning of Life and SES of Undergraduates*

The average value of undergraduates' sense of meaning in life is 5.04. Specifically, the average value of meaning experience is 4.60, and the average value of the meaning seeking dimension is 5.47. Considering that the median value of the undergraduates' sense of meaning in life scale is 4, the average value of the total score of the undergraduates' sense of meaning in life was slightly higher than 4. Therefore, it can be said that the undergraduates' sense of meaning in life is at a moderate level in general.

The average education level of undergraduates' fathers is 1.78 (1 = junior high school and below, 2 = high school), and the average education level of mothers is 1.59, which means that the average education level of undergraduates' parents is high school and below. The average monthly household income is 2.10 (1 = less than CYN 4000, 2 = CYN 4001–6000, and 3 = CYN 6001–8000), which means that the average monthly household income is around CYN 6000. The average objective SES of undergraduates is 1.83; the average current SES of undergraduates is 3.35; and the average future SES is 7.80. The paired *t*-test between current SES and future SES showed that undergraduates' current SES was significantly lower than future SES, $t_{(332)} = -36.85$, $p < 0.001$ (see Figure 1).

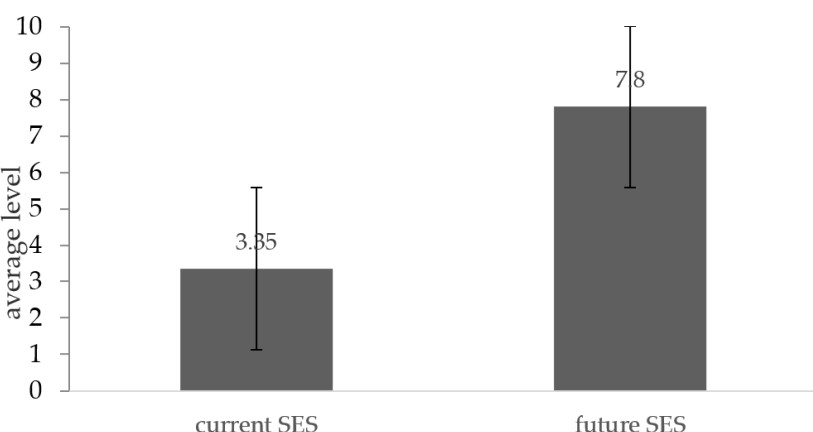

**Figure 1.** Average level of undergraduates' current SES and future SES.

To test whether there was a difference in the sense of meaning in life and SES between males and females, an independent sample *t*-test or a Mann–Whitney U test was conducted. The results are shown in Table 1. The total score for the meaning of life for males (*M* = 5.12, *SD* = 0.93) was significantly higher than that of females (*M* = 4.92, *SD* = 0.80). The meaning experience of males (*M* = 4.76, *SD* = 1.24) was significantly higher than that of females (*M* = 4.39, *SD* = 1.21). The future SES of males (*M* = 7.89, *SD* = 2.04) was significantly higher than that of females (*M* = 7.67, *SD* = 1.60). The mother's education of males (*M* = 1.50, *SD* = 0.90) was significantly lower than that of females (*M* = 1.71, *SD* = 0.95). There were no significant sex differences in meaning seeking, current SES, the father's education level, family income, or objective SES.

**Table 1.** The descriptive analysis of undergraduates' sense of life meaning and SES ($M \pm SD$).

| | | Sex | | | | Rural/Urban | | |
| --- | --- | --- | --- | --- | --- | --- | --- | --- |
| | Male | Female | *t*/*Z* | *p* | Rural | Urban | *t*/*Z* | *p* |
| 1 | 5.12 ± 0.93 | 4.92 ± 0.80 | 2.01 * | 0.04 | 5.00 ± 0.88 | 5.12 ± 0.87 | −1.07 | 0.29 |
| 2 | 4.76 ± 1.24 | 4.39 ± 1.21 | 2.68 ** | 0.01 | 4.59 ± 1.22 | 4.63 ± 1.29 | −0.24 | 0.81 |
| 3 | 5.48 ± 1.05 | 5.46 ± 1.00 | 0.43 | 0.66 | 5.42 ± 1.04 | 5.61 ± 0.99 | −1.61 | 0.11 |
| 4 | 3.30 ± 1.69 | 3.42 ± 1.70 | −0.58 | 0.56 | 3.15 ± 1.61 | 3.84 ± 1.79 | −3.48 ** | 0.001 |
| 5 | 7.89 ± 2.04 | 7.67 ± 1.60 | 2.09 * | 0.04 | 7.77 ± 1.94 | 7.85 ± 1.66 | −0.17 | 0.87 |
| 6 | 1.73 ± 1.05 | 1.84 ± 1.02 | −1.43 | 0.15 | 1.45 ± 0.82 | 2.61 ± 1.05 | −9.65 ** | 0.001 |
| 7 | 1.50 ± 0.90 | 1.71 ± 0.95 | −2.42 * | 0.02 | 1.28 ± 0.65 | 2.36 ± 1.07 | −9.77 ** | 0.001 |
| 8 | 2.14 ± 1.19 | 2.04 ± 1.03 | 0.64 | 0.52 | 1.86 ± 1.02 | 2.72 ± 1.28 | −5.84 ** | 0.001 |
| 9 | 1.79 ± 0.82 | 1.87 ± 0.83 | −1.12 | 0.26 | 1.53 ± 0.59 | 2.56 ± 0.87 | −9.91 ** | 0.001 |

Note: * $p < 0.05$, ** $p < 0.01$. 1 = total meaning in life, 2 = meaning experience, 3 = meaning seeking, 4 = current SES, 5 = future SES, 6 = father's education, 7 = mother's education, 8 = monthly household income, 9 = objective SES. For variables 3–9, Mann–Whitney *U* test instead of *t*-test was conducted due to the non-normal distribution.

In order to explore the differences in the sense of meaning in life and SES between students from urban and rural areas, the independent sample *t*-test or Mann–Whitney U test was conducted. We found that urban students' current SES, father's education, mother's education, monthly household income, and objective SES were higher than rural students', while there were no significant differences between rural and urban students for the total meaning in life, the dimension of meaning experience, the dimension of meaning seeking, and future SES.

### 3.2. The Relationship between Undergraduates' SES and Sense of Meaning in Life

The correlation analysis between undergraduates' SES and sense of meaning in life is shown in Table 2. It can be seen that there was no significant correlation between objective SES and undergraduates' meaning experiences, meaning seeking, and total score of sense of meaning in life. Current SES is significantly positively correlated with undergraduates' meaning experience ($r = 0.12$, $p < 0.05$), but there was no significant correlation either with meaning seeking ($r = 0.02$, $p > 0.05$) or with the total score of sense of meaning in life ($r = 0.09$, $p > 0.05$). Future SES was positively correlated with undergraduates' meaning experience ($r = 0.21$, $p < 0.001$), meaning seeking ($r = 0.12$, $p < 0.05$), and total score of meaning in life ($r = 0.22$, $p < 0.001$).

**Table 2.** Correlation analysis between socioeconomic status and sense of meaning in life.

| | 1 | 2 | 3 | 4 | 5 | 6 |
| --- | --- | --- | --- | --- | --- | --- |
| 1 Total meaning in life | 1 | | | | | |
| 2 Meaning experience | 0.82 *** | 1 | | | | |
| 3 Meaning seeking | 0.72 *** | 0.19 *** | 1 | | | |
| 4 Objective SES | 0.01 | −0.01 | 0.02 | 1 | | |
| 5 Current SES | 0.09 | 0.12 * | 0.02 | 0.21 *** | 1 | |
| 6 Future SES | 0.22 *** | 0.21 *** | 0.12 * | 0.04 | 0.24 *** | 1 |

Note: * $p < 0.05$, *** $p < 0.001$.

In order to further examine the impact of future SES on undergraduates' sense of meaning in life on the basis of controlling for sex, age, objective SES, and current SES, regression analysis was conducted with future SES as an independent variable and the meaning experience, meaning seeking, and total score of sense of meaning as dependent variables, respectively. The results are shown in Table 3. First, it can be seen that after controlling for the effects of sex, age, objective SES, and current SES, the future SES has a significant impact on the meaning experience ($\beta = 0.17$, $p < 0.01$), meaning seeking ($\beta = 0.13$, $p < 0.05$), and the total score of meaning in life ($\beta = 0.20$, $p < 0.01$), while neither objective

SES nor current SES predicted the meaning of life significantly. This result showed that the future SES significantly predicted the meaning of life for undergraduates.

**Table 3.** Regression analysis of undergraduates' sense of meaning in life on SES.

| | Meaning Experience ($\beta$) | | Meaning Seeking ($\beta$) | | Total Score of Meaning in Life ($\beta$) | |
|---|---|---|---|---|---|---|
| sex | 0.13 * | 0.12 * | 0.02 | 0.01 | 0.10 | 0.09 |
| age | 0.15 ** | 0.14 ** | −0.06 | −0.07 | 0.07 | 0.06 |
| objective SES | −0.03 | −0.02 | 0.02 | 0.02 | −0.01 | −0.09 |
| current SES | 0.13 * | 0.09 | 0.01 | −0.02 | 0.10 | 0.92 |
| future SES | | 0.17 ** | | 0.13 * | | 0.20 *** |
| $F$ | 5.22 *** | 6.32 *** | 0.32 | 1.23 | 2.25 | 4.36 ** |
| $R^2$ | 0.06 | 0.09 | 0.004 | 0.02 | 0.0 3 | 0.07 |
| $\Delta R^2$ | | 0.03 ** | | 0.02 * | | 0.04 ** |

Note: * $p < 0.05$, ** $p < 0.01$, *** $p < 0.001$. Sex is a dummy variable, 1 = male, 0 = female.

### 3.3. The Mediating Role of the Satisfaction of Basic Psychological Needs

In order to test whether the satisfaction of basic psychological needs plays a mediating role in the impact of future SES on undergraduates' sense of meaning in life, this study conducted a three-step regression analysis, and the results are shown in Table 4. In the first step, after controlling for sex, age, objective SES, and current SES, future SES can significantly predict the sense of meaning in life ($\beta = 0.20$, $p < 0.001$). In the second step, future SES can significantly predict the satisfaction of basic psychological needs ($\beta = 0.21$, $p < 0.001$), after controlling sex, age, objective SES, and current SES. In the third step, both future SES and basic psychological needs satisfaction entered the regression analysis, and the results showed that future SES ($\beta = 0.11$, $p < 0.001$) and basic psychological needs satisfaction ($\beta = 0.43$, $p < 0.001$) can significantly predict the sense of meaning in the lives of undergraduates, indicating that the satisfaction of basic psychological needs played a partial mediating role in the impact of future SES on the sense of meaning in the lives of undergraduates, see Figure 2.

**Table 4.** Analysis of the mediating effect of basic psychological needs satisfaction between future SES and undergraduates' sense of meaning in life.

| Independent Variable | Step 1: Sense of Meaning in Life | Step 2: Basic Psychological Needs Satisfaction | Step 3: Sense of Meaning in Life |
|---|---|---|---|
| sex | 0.09 | −0.09 | 0.13 * |
| age | 0.06 | 0.08 | 0.03 |
| objective SES | −0.09 | 0.07 | −0.04 |
| current SES | 0.92 | 0.01 | 0.05 |
| future SES | 0.20 *** | 0.21 *** | 0.11 * |
| basic psychological needs | | | 0.43 *** |
| $F$ | 4.36 ** | 4.61 *** | 16.53 *** |
| $R^2$ | 0.07 | 0.06 | 0.23 |

Note: * $p < 0.05$, ** $p < 0.01$, *** $p < 0.001$. Sex is a dummy variable, 1 = male, 0 = female.

In order to further examine the mediating effects of the three basic psychological needs, i.e., competence need, relatedness need, and autonomy need, respectively, we used the Bootstrap method to test the mediating effect while controlling sex, age, objective SES, and current SES. The results were shown in Table 5. The indirect effect of competency need satisfaction was 0.08 with a 95% confidence interval of [0.0314, 0.1397], excluding 0, indicating that the mediating effect of competency need satisfaction was significant. The indirect effect of relatedness need satisfaction was 0.06, 95% confidence interval [0.0014, 0.0572], excluding 0, indicating that the mediating effect of relatedness need satisfaction was significant. The indirect effect of autonomy need satisfaction was 0.01, with a 95%

confidence interval [−0.0160, 0.0266], including 0, indicating that the mediating effect of autonomy need satisfaction was not significant. In other words, the mediation of competence need satisfaction and relatedness need satisfaction were significant, but the mediation of autonomy need satisfaction was not significant.

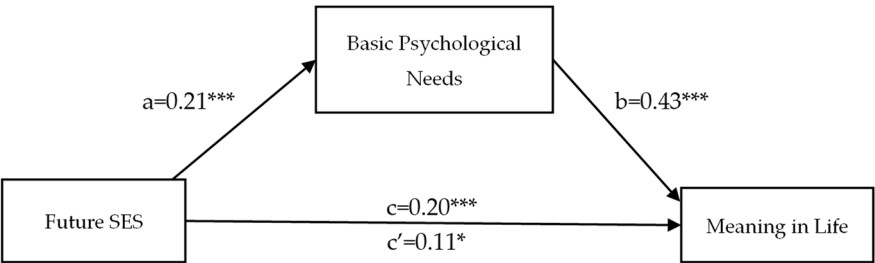

**Figure 2.** The mediating effect of basic psychological needs in the relationship between future SES and meaning in life. Note: * $p < 0.05$, *** $p < 0.001$.

**Table 5.** The mediating effect of different basic psychological needs satisfaction in the relationship between future SES and undergraduates' sense of meaning in life.

| Mediator | Total Effect | Direct Effect | Indirect Effect | Boot SE | Bootstrap 95% CI | |
|---|---|---|---|---|---|---|
| competency need satisfaction | 0.46 | 0.38 | 0.08 | 0.03 | 0.0314 | 0.1397 |
| relatedness need satisfaction | 0.46 | 0.40 | 0.06 | 0.01 | 0.0014 | 0.0572 |
| autonomy need satisfaction | 0.46 | 0.45 | 0.01 | 0.01 | −0.0160 | 0.0266 |

## 4. General Discussion

The current study explored the influence of future SES on the sense of meaning in life of undergraduates and the mediation of basic psychological needs satisfaction in the effect of future SES on meaning in life.

The result of descriptive analysis showed that currently undergraduates' meaning in life was 5.04, which suggested that the overall level of undergraduates' meaning in life is at a moderate level in general, which is a little higher than that of medical students (Wang et al. 2018). The average future SES was significantly higher than the current SES. The future SES of males was significantly higher than that of females, while for the current SES, there was no significant difference between male and female undergraduates. There was no significant difference between the future SES of rural and urban areas, while the current SES of urban areas was significantly higher than that of rural areas. The results also showed that the total meaning of life and experience of meaning of males were significantly higher than those of females. However, for the meaning of life, there was no significant difference between undergraduates from rural and urban areas.

The result of the correlation analysis indicated that future SES was positively related to undergraduates' sense of meaning in life, which supports Hypothesis 1. Previous studies also found that subjective SES was positively related to meaning in life (Shang et al. 2016; Tang et al. 2021; Zhao et al. 2019; Shek 2003), which was consistent with the present results.

Mediation analysis found that basic psychological needs satisfaction mediated the effect of future SES on undergraduates' sense of meaning in life, which supported Hypothesis 2. Further, the mediating effects of the three basic psychological needs—competence need, relatedness need, and autonomy need—were examined, respectively, and results showed that the mediating effects of relatedness need satisfaction and competence need satisfaction were significant, but the mediating effect of autonomy need satisfaction was not significant. In other words, undergraduates expecting high SES in the future will improve

their sense of meaning in life by satisfying their competency needs and relatedness needs rather than their autonomy needs. A previous study provided evidence that psychological need satisfaction contributed to the sense of meaningfulness (Martela et al. 2018). If the psychological needs were not met, the individual would have a sense of uncertainty, insecurity, self-rejection, and rejection (Deci and Ryan 2000, 2008; Yang et al. 2017). Previous studies also found that satisfying basic psychological needs enhanced the sense of meaning in life (Zhang et al. 2022; Eakman 2014; Yalçın 2022), and if the competency need was met, people would experience a stronger sense of meaning in life even in difficult environments (Huynh and Chiang 2018).

The sense of meaning in life has long been a mystery of human existence. Previous studies have proven that one's sense of meaning in life is a key predictor of mental health (Klinger 1977; Park 2005; Stroope et al. 2013). The current study makes a contribution to this research field by demonstrating the impact of future SES on the sense of meaning in life. Studies have mentioned that prospection, or thinking about one's self in the future, has an impact on the sense of meaning in life (Baumeister et al. 2016; van Tilburg and Igou 2019). It is demonstrated that participants considered thoughts about the future to be more meaningful than thoughts about the present (Baumeister and Hippel 2020). Other studies proved that imagining a meaningful event in the future (i.e., turning a new decade in age) led people to reflect over their entire lives (Kim et al. 2019). The present research suggests that thinking positively about future SES has the potential to feel meaningful because it often leads to a feeling of coherence among varied aspects of life. Therefore, the present study contributed to the research field of meaning in life by examining the impact of future SES on undergraduates' sense of meaning in life.

The present study also contributed to a better understanding of the psychological underpinnings of the association between future SES and undergraduates' sense of meaning in life. Psychological need satisfaction is often regarded as a source of well-being and meaning in life (Martela and Sheldon 2019; Shin and Park 2022; Lataster et al. 2022). We accessed the relationship between future SES and a sense of meaning in life by emphasizing the importance of psychological need satisfaction.

For practical implications, it can provide effective suggestions to improve undergraduates' sense of meaning in life. For example, colleges and universities could help undergraduates build positive future career plans, gradually exercise and improve students' competence, and guide students to build harmonious interpersonal relationships, which will help students meet their basic psychological needs better, thus finally improving their sense of meaning in life.

A few limitations of the study are worth mentioning. First, this study was a cross-sectional study, so future research could validate the results of this study through longitudinal studies. Second, this study only examined the impact of the future SES on undergraduates' sense of meaning in life. Therefore, further studies are needed to explore other variables that may have an effect. Third, the sample size needs to be further expanded. Fourth, in this study, undergraduates' meaning in life is based on self-reports, which are susceptible to variable biases such as social desirability (Delroy 1991). Thus, social desirability should be considered in future research.

## 5. Conclusions

In conclusion, the present study provides evidence showing that undergraduates' sense of meaning in life was at the middle level in general. Their current SES was low, but their future SES was high. Future SES was positively related to undergraduates' sense of meaning in life. Psychological need satisfaction plays a mediating role between future SES and undergraduates' sense of meaning in life. In specific, it is the satisfaction of competence and relatedness needs, rather than the satisfaction of autonomy needs, that mediated the effect of future SES on meaning in life. In other words, undergraduates with higher expectations of their future SES can meet their need for competence and relatedness better and then have a stronger sense of meaning in life. Based on these results, there

are a number of implications for college students' meaning in life and mental health in China. Targeted and effective mental health education activities could be used to encourage undergraduates to pursue the meaning of life. Teachers and counselors could provide career counseling to undergraduates to enable them to have a clear blueprint and plan for their future, thus improving their confidence in their future SES, which then leads to a stronger sense of meaning in life. In other words, the present research provided a new effective tool to increase undergraduates' meaning in life, i.e., increasing their future SES and satisfying their basic psychological needs.

**Author Contributions:** Conceptualization, X.G. and F.Z.; methodology, L.S. and X.G.; writing—original draft preparation, L.S. and F.Z.; writing—review and editing, X.G.; funding acquisition, F.Z. All authors have read and agreed to the published version of the manuscript.

**Funding:** This research was funded by the Social Science Planning Research Project in Shandong Province (18CJYJ13).

**Institutional Review Board Statement:** The study was conducted according to the guidelines of the Declaration of Helsinki, and it was approved by the Academic Ethics Committee of Hangzhou Normal University, approval code 2023008.

**Informed Consent Statement:** Informed consent was obtained from all participants involved in the study.

**Data Availability Statement:** According to the data access policies, the data used to support the findings of this study are available upon a reasonable request made by email: xwgeng@hznu.edu.cn.

**Conflicts of Interest:** The authors declare no conflict of interest.

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
