# Peer review of "The Mediating Effect of Basic Psychological Needs Satisfaction between Future Socio-Economic Status and Undergraduates’ Sense of Meaning in Life"

_socsci, doi:10.3390/socsci12040229_

Round 1
Reviewer 1 Report
Dear Authors,
I received a very interesting article for review. However, the paper needs some revisions so that it can be well appreciated by the intended audience.
The title of this article should be rethought because it does not present the content of the article - it must be changed and clarified.
The entire article should be adapted to the Template of “IJERPH”! IT’S VERY IMPORTANT.
There is no clear link between the title of the article, objectives, research
methods and the summary and the conclusions - to delete!
The research results should be clarified in the conclusions or the discussion
BUT
General discussion can be accepted provided that the Authors expand, correct, and ideally rewrite the conclusions. According to the Reviewer, the current version of the conclusions is unacceptable. The conclusions are so unacceptable that they could affect the "reject" rating of the reviewed article.In my opinion, the number of cited sources is too small, because it reduces the substantive value of the article.
Good Luck,
Reviewer
Reviewer 2 Report
Dear researchers, first of all, congratulations to you on this research, it is an interesting and relevant topic in a university context. After reading this document, I had the following doubts about the paper:
1. Introduction. · Lines 23 and 24, what would be the references? · References 1 and 2, one of them is from 2010 and the other from 2016. How much literature currently exists on this topic? On the other hand, it is said that 50% of undergraduates..., for this statement a more solid reference and I think that two references that are more than 5 years old (one of them is from 2010) can support this statement. · Lines 26, 27 and 28, Who is the author who supports this statement? From what paradigm is it approached? From the philosophical, education, health, for example. · Why is there talk of a subjective SES? The definition is not entirely clear. · Is it necessary to put the phrase in bold? General comments from the introduction:
Dear researchers, the introduction that you develop has several statements without the corresponding bibliographic support. On the other hand, the structure that you present is a bit abstract or ambiguous. If you do a brief search in Pubmed (for example) perhaps you could address "the meaning of life" in a better way, although it is a multifactorial issue or many causes converge on this issue, I think ordering the introduction will be kinder to understand what they really developed or are looking for.
Another not minor point, if clear guidelines are not established in the introduction, this work could take another direction, for example we talk about well-being, but this term is broad, what do you hope to find, a contribution of the biopsychosocial human being (this is a Engel's proposal from 1973, but there are authors who, mentioning more dimensions of the human being, from 5 to 12 dimensions, among them are labor, economic, spiritual, among others, we are in the year 2023 and there are many interesting proposals)?
On the other hand, and an important issue, there is talk of the subjective SES, but it can only be understood when reading the methodology, it is understood that it is a "MacArthur Scale" instrument, which, by the way, I think the quote is not consistent to the magazine standard. Also, you declare that based on 23 articles (literature) and some statements without reference, 2 hypotheses are proposed.
Finally, I invite you to restructure your introduction, and be clearer about what you are looking for, beyond a clear objective, I believe that the introduction is a letter of entry when presenting a work, of course, from my point of view as This is how I experienced it professionally when sending my research to journals.
2.- Materials and methods
These 333 subjects, were they selected for convenience? What major were they studying? What instrument for the target socioeconomic level did they use? Did they live in an urban or rural area? What education did their parents have? You mention that the instrument was widely used, can cite some papers.
Regarding the instruments, were they previously validated for the sample to be selected?https://bmcpublichealth.biomedcentral.com/articles/10.1186/1471-2458-12-1096)
The treatment of the data is not entirely clear. Which variables were used with which variables?
3.- Results
Why isn't there a table with descriptive characteristics? Why isn't there information from a table on objective socioeconomic data? What are the differences between objective and subjective socioeconomic data? I think this section deserves better development and more clear. Each question has only been described in a general way, but how are the results evidenced by questions between men and women? Is there any difference considering objective data? “and” do they not perceive differently? For example, in some countries they take admission tests to universities (to study a degree), if you do not have the corresponding qualification and you study another degree that you can study due to your grade, how do you perceive that future SES, what predisposition do students have to answer the survey?, this could be considered bias.
Point 3.1 of the results, What questions are in this instrument? It is necessary to present a table with the questions (perhaps shorten the original questions and present them in the results, the original instrument can be attached as annexes), the educational level of parents is 1.78, but basic, incomplete, university?
Is it possible to be a little clearer with the variables used in mediation?, for example:
https://www.mdpi.com/1660-4601/20/4/3160
https://www.mdpi.com/1660-4601/20/4/3105
Is the target economic level divided into yuan, 4000, 4001 – 6000, 6001 – 7000 is low, medium and high?, students would have an average SES of 2.1, and the future SES would be 7.80?, I think a visual aid It would allow a better understanding. Is this comparison correct?
4.- Discussion.
I think the discussion is very weak, on the other hand, there are no strengths, and finally they only consider "two limitations", are you sure? I have visualized at least 4 more limitations.
5. Conclusions I think it is not clear. Others: · Why ** University appears in the ethical aspect? · Also consider recent articles. · Increase the number of references.
Reviewer 3 Report
The research is ingeniously conducted by measuring SES through 3 dimensions: future, objective and subjective and then the relationship with meaning in life and basic psychological needs.
Line 128 and Line 141: it is understood that Cronbach's alpha is for the total score. Let this be mentioned.
MLQ and BPNS: versions validated on the Chinese population were used
Regarding the fact that a t-test was used, did the data present a normal distribution? Mention what the situation is.
Results
Line 240: theoretically, the total and direct effects had to be expressed as well
Minor aspect:
Line 99: conversion of journal-specific citation
Round 2
Reviewer 1 Report
Dear Authors,
thank you for your reply and cover letter.
I accept this version of the article, but please read the
full text once again and proofread the text.
In my opinion, References are at the moment too small (only 48).
Good Luck,
Reviewer
Reviewer 2 Report
Dear researchers,
Once again, congratulate you on your work, it has clearly undergone substantial changes. However, I have some doubts about the new changes made:
Abstract:
Why was the word "subjective" changed to "current"?
Introduction:
· I couldn't find the papers for line 28, is citation 2 a book chapter?
· The papers in line 30 are extensive, they speak of various fields, is this part of your research?
· Ref. 11 is it a book?, because this ref. defines the concept that is being used.
· Lines 45 and 46, they talk about SES Objective and Subjective, but the summary mentions "current", why was this change made?
Methodology:
· Information is detailed in a better way
Results:
· Review the tables and figures please. There are some that do not contain information at the bottom of them (or at the top), and table 4 has words in black.
· I have doubts, what is the difference between "subjective" and "current"
Discussion:
· I only have doubts with the references that have been used.
Conclusion:
· It has improved, it has presented itself in an improved way.
References
· I had small problems, I couldn't access some references, perhaps because the journal is very, very specific, such as papers 1 and 2, among others.
